# The Potentiating Effect of Graphene Oxide on the Arylhydrocarbon Receptor (AhR)–Cytochrome P4501A (Cyp1A) System Activated by Benzo(k)fluoranthene (BkF) in Rainbow Trout Cell Line

**DOI:** 10.3390/nano13182501

**Published:** 2023-09-05

**Authors:** Ana Valdehita, María Luisa Fernández-Cruz, José M. Navas

**Affiliations:** Instituto Nacional de Investigación y Tecnología Agraria y Alimentaria (INIA), CSIC, Carretera de la Coruña Km 7,5, E-28040 Madrid, Spain; ana.valdehita@inia.csic.es (A.V.); fcruz@inia.csic.es (M.L.F.-C.)

**Keywords:** graphene, fish, EROD, aryl hydrocarbon receptor, Cyp1A

## Abstract

The increasing use of graphene oxide (GO) will result in its release into the environment; therefore, it is essential to determine its final fate and possible metabolism by organisms. The objective of this study was to assess the possible role of the aryl hydrocarbon receptor (AhR)-dependent cytochrome P4501A (Cyp1A) detoxification activities on the catabolism of GO. Our hypothesis is that GO cannot initially interact with the AhR, but that after an initial degradation caused by other mechanisms, small fractions of GO could activate the AhR, inducing Cyp1A. The environmental pollutant benzo(k)fluoranthene (BkF) was used for the initial activation of the AhR in the rainbow trout (*Oncorhynchus mykiss*) cell line RTL-W1. Pre-, co-, and post-exposure experiments with GO were performed and Cyp1A induction was monitored. The strong stimulation of Cyp1A observed in cells after exposure to GO, when BkF levels were not detected in the system, suggests a direct action of GO. The role of the AhR was confirmed by a blockage of the observed effects in co-treatment experiments with αNF (an AhR antagonist). These results suggest a possible role for the AhR and Cyp1A system in the cellular metabolism of GO and that GO could modulate the toxicity of environmental pollutants.

## 1. Introduction

Since the discovery of graphene in 2004 [1], few manufactured nanomaterials (MNs) have received such rapid and great attention from science. Ideally, graphene is constituted of carbon atoms with a sp2 hybridization that forms a lattice that could be represented in a similar way to a single-layer honeycomb structure. However, in most of the cases, graphene does not appear as such in a reduced form but contains different functional groups (e.g., hydroxyl, carboxyl, and epoxy groups) and is denoted as graphene oxide (GO). These functional groups facilitate its suspension in water or organic solvents as well as its interaction with different biomolecules [2]. As a consequence, biomedical applications for GO have been found, for instance, in diagnostics, targeted drug delivery, and even gene therapy [3,4,5]. GO can also be applied in environmental remediation, where it is used for ultrafiltration, photocatalysis, and sewage treatment [6,7,8]. The increasing production and use of GO will inevitably lead to its release into the aquatic environment, which has been an increasing cause of concern in recent years [9,10,11,12]. Therefore, the highly interactive nature of GO with living systems, either in a controlled way during its use and application or accidentally after its release, makes it essential to gain knowledge about its potential toxicity and its possible metabolization by organisms.

Most toxicological studies on GO have addressed its potential toxicity to humans due to its mentioned potential use in biomedicine [13,14,15]. There exists, however, a number of studies concerning the adverse effects of GO on aquatic species such as bacteria, algae, and fish [11,15,16,17,18,19,20]. The underlying mechanisms of toxicity reported for GO in fish have been mainly associated with the generation of reactive oxygen species (ROS) and the induction of oxidative stress [20].

Although several recent studies have shown the toxicity of GO to aquatic species, reviewed by Connolly et al. [21], very little is known about the adverse effects of GO when it interacts with other co-occurring pollutants in aquatic ecosystems [10,22]. GO has an outstanding capacity to interact with or adsorb organic pollutants such as polycyclic aromatic hydrocarbons (PAHs) and polychlorinated biphenyls (PCBs) [23]. However, to the best of our knowledge, few toxicity studies considering the possible mutual influence of graphene derivatives and environmental toxicants have been carried out [24].

In vitro assays using fish cell lines can be a powerful tool to determine the toxicity of MNs and graphene-related materials (GRMs) [25,26,27]. In vitro approaches can be used in a preliminary screening of toxicity to determine the most suitable concentrations to be used in higher-tier in vivo tests, and, additionally, are an essential tool for unraveling the mechanisms of toxic action [28,29]. In a number of previous works, we have shown that GO was able to pierce the plasma membrane and accumulate in the cytosol of a variety of fish cell lines, decreasing their viability [25,27,30]. We also observed, for the first time [24] while using a fish cell line, that GO had a strong potentiating effect on the induction caused by aromatic environmental pollutants on a cytochrome with a key role in cellular detoxification processes, cytochrome P450 1A (Cyp1A). Later, other studies showed an up-regulation of Cyp1A caused by GO, specifically GO quantum dots [31,32,33]. Cyp1A plays a key role in the biotransformation of a wide variety of xenobiotics and, therefore, has been widely used as a biomarker of environmental pollution in aquatic ecosystems [34,35] by measuring the associated enzyme activities such as ethoxyresorufin-*O*-deethylase (EROD). The induction of CypP1A occurs after the ligand activation of the aryl hydrocarbon receptor (AhR). This receptor is typically, but not exclusively, activated by planar, polycyclic, and aromatic compounds, including some polyaromatic hydrocarbons (PAHs) or dioxins [36]. Therefore, considering that the graphene structure could be assimilated to that of aromatic hydrocarbons, the question is raised about the possible role of AhR and Cyp1A on the metabolism of graphene.

To deepen our understanding of these issues, the present work was performed with the aim of observing a possible role of Cyp1A-dependent detoxification activities on GO metabolism. The hypothesis is that GO cannot initially interact with the AhR and induce the expression of Cyp1A, but that after an initial intracellular degradation by other mechanisms, fractions of GO could be able to provoke the induction of Cyp1A. In the present study, the initial activation of the AhR was achieved by exposure to the environmental pollutant and prototypical AhR ligand, benzo(k)fluoranthene (BkF). To determine the best exposure conditions, pre-, co-, and post-exposure experiments with BkF and GO were carried out. In the post-exposure experiments (24 h with BkF, 7 days with GO) a clear potentiation of Cyp1A-dependent activities was observed by GO. Since BkF has also the ability to trigger other detoxification responses, including the expression of Cyp3A [37], the induction of this Cyp was also assessed at the transcriptional and enzyme activity (by measuring Benzyloxy-4-trifluoromethylcoumarin-*O*-debenzyloxylase, BFCOD) levels for comparative purposes.

Routinely, only short-term (2–24 h) toxicity experiments are carried out on cell cultures, but we considered that longer-term experiments of 7 days of exposure could give essential information about the mechanisms underlying the observed effects. To facilitate this, we used a rainbow trout liver-derived cell line, RTL-W1, both due to the key role played by the liver in detoxification processes and this cell line’s robustness.

## 2. Materials and Methods

### 2.1. Chemicals

All chemicals were purchased from Sigma Aldrich (Madrid, Spain) unless otherwise stated. For the cell culture, L-Glutamine (200 mM), penicillin and streptomycin (P/S) (10,000 U/mL each), trypsin-ethylenediamine tetraacetic acid (EDTA) (200 mg/mL), a non-essential amino acid (NEAA) solution, and Leibovitz’s (L-15) cell culture medium were purchased from Lonza (Barcelona, Spain). Phenol red-free serum-free Minimum Essential Medium (MEM) was supplied by Gibco (Life Technologies, Madrid, Spain). Among the reactants used for determining cytotoxicity, resazurin (AlamarBlue^®^, AB) and 5-carboxyfluorescein diacetate acetoxy methyl ester (CFDA-AM) were purchased from Invitrogen (Madrid, Spain). For the electron microscopy analyses, paraformaldehyde (16%) and glutaraldehyde (25%) were supplied by Electron Microscopy Sciences (Hatfiled, UK), and Spurr´s resin was provided by TAAB Laboratories Equipment Ltd. (Aldermaston, UK). High-grade purity water (>18 MΩ cm^−1^) was obtained from a Milli-Q ElementA10 Century (Millipore Iberia, Madrid, Spain).

### 2.2. Graphene Oxide Suspension Preparation and Characterization

Graphene oxide was synthesized and provided in powder form (GRAnPH^®^) by Grupo Antolín Ingeniería, S.A. (Burgos, Spain). According to the manufacturer and as described in previous work [25], GRAnPH^®^ consists of a single or few layers of graphene sheets and exhibits a lateral size < 1 µm. The GO concentration was ≥100%, so no impurities were expected. In the initial experiments (see below), GO suspensions did not provoke any detoxification activity induction, so the presence of possible interfering impurities was discarded. The same batch was used throughout the study to rule out variations due to the use of different batches.

GO stock dispersions were prepared as described in Lammel et al. [30]. In brief, GO was dispersed at 1 mg/mL in Milli-Q water using an ultrasonic bath at 50–60 Hz for 30 min (stock suspension). The resulting suspensions were centrifuged at 1300× *g* for 30 min to remove large aggregates/agglomerates. The concentrations of the supernatants of the GO suspensions were determined using concentration/absorbance standard curves as previously described [30] and then adjusted to 750 µg/mL (stock suspension after centrifugation). The hydrodynamic diameters (HDD) and size–frequency distribution of this stock suspension were measured by means of dynamic light scattering (DLS) using a Zetasizer Nano Series (Malvern Instruments, Malvern, UK) at time 0 and after 7 days. For cell treatment, GO suspensions were diluted 1:10 in a cell culture medium, resulting in a maximum exposure concentration of 75 µg/mL. The HDD and size frequency distribution were measured at 4.16, 18.75, and 75 μg/mL of GO at different time intervals (0 and 7 days). All suspensions of GO were freshly prepared before each in vitro toxicity assay. Three to ten independent measurements were carried out, with each measurement consisting of six runs. GO’s morphology was also assessed by means of transmission electron microscopy (TEM). Samples were prepared by placing a drop of the suspensions onto carbon-coated copper TEM grids and allowed to evaporate at room temperature before analysis. TEM analysis was carried out using a JEOL 1400 Plus TEM (JEOL Ltd., Tokyo, Japan).

### 2.3. Cell Culture

The RTL-W1 rainbow trout (*Oncorhynchus mykiss*) cell line was a generous gift from Drs. Lee and Bols, who obtained it from normal liver tissue [38]. Cells were cultured in 75 cm^2^ CellStar cell culture flasks (Greiner Bio-One GmbH, Germany) in Leibovitz’s L-15 culture medium supplemented with 10% fetal bovine serum (FBS), 1% penicillin/streptomycin (P/S) solution, and 1% L-Glutamine. Flasks were incubated at 20 °C and split twice a week using PBS/EDTA and Trypsin-EDTA to detach cells from the bottom of the flasks.

### 2.4. Internalization of GO

Transmission electron microscopy was used to investigate the internalization of GO into the RTL-W1 cells. For that, cells were seeded on poly-L-lysine coated coverslips in a 24-well plate (1.0 × 10^5^ cells/well) and exposed to 4.6 and 18.75 µg/mL of GO for 7 days. After exposures, sample preparation was performed as described by Lammel et al. [30] including washing steps (Millonig’s phosphate buffer, pH 7.3), primary fixation (4% paraformaldehyde 2.5% glutaraldehyde), post fixation (1% osmium tetroxide), gradual dehydration steps (30–100% acetone), embedding (gradual infiltration with Spurr’s resin), and a polymerization step (65 °C, 48 h). Ultrathin sections were stained in uranyl acetate and lead citrate and observed in a JEOL 1010 JEM TEM (JEOL Ltd., Tokyo, Japan).

### 2.5. RTL-W1 Cell Treatment

RTL-W1 cells were seeded into 96-well plates at a concentration of 2.5 × 10^5^ cells/mL (100 µL/well). They were allowed to attach to the bottom of the wells for 24 h. Different exposure experiments in co-incubation, pre-incubation, and post-incubation were carried out as described below.

#### 2.5.1. Single Exposure Experiments

RTL-W1 cells were treated with concentrations of GO ranging between 0.3 and 75 µg/mL (using a dilution factor of 2) or with an AhR agonist, BkF, at a wide range of concentrations, from 0.01 nM to 1 mM for 7 days, in a final volume of 200 µL. Although environmentally relevant concentrations (0.050 to 0.1 µg/mL) [39] are lower than those used in this work (0.3–75 µg/mL), the applied GO concentrations were selected according to the criteria set out in OECD Test Guideline nº 249 which states that the highest concentration tested should preferably result in 0% cell viability compared to the solvent control. The lowest test chemical concentration should preferably give no effect and therefore will result in 100% cell viability compared to the solvent control [40]. Control wells were treated with medium or medium plus the maximal solvent concentration: 10% *v*/*v* MilliQ water/medium for GO or a maximum of 0.01% DMSO for BkF.

#### 2.5.2. Co-Exposure Experiments

RTL-W1 cells were treated with increasing concentrations of BkF (0.008–1 µM, dilution factor of two) alone or in the presence of 4.7 and18.75 µg/mL of GO in a final volume of 200 µL. Cells treated with 0.01% DMSO, 0.6 and 2.5% (*v*/*v*) Milli-Q water/medium served as vehicle controls.

#### 2.5.3. GO Pre-Exposure Experiments

RTL-W1 cells were pre-exposed to 4.7 and 18.75 µg/mL GO for 7 days, and cells treated with 0.6 and 2.5% (*v*/*v*) Milli-Q water served as vehicle controls. Thereafter, the culture medium was aspirated, and the cells were rinsed twice with PBS. Then, the cells were incubated with increasing concentrations of BkF (0.008–1 µM) for 24 h. Cells treated with 0.01% DMSO served as vehicle controls.

#### 2.5.4. GO Post-Exposure Experiments

RTL-W1 cells were exposed to increasing concentrations of BkF (0.008–1 µM) or 0.01% DMSO as a negative control for 24 h, then rinsed twice with PBS and post-exposed to 4.7 and 18.75 µg/mL GO for 7 days or with 0.6 and 2.5% (*v*/*v*) Milli-Q water (cells used as vehicle controls).

### 2.6. Cytotoxicity Assays

Cell viability was assessed through three different assays, AB, CFDA-AM, and Neutral Red uptake (NR), which give information about the possible changes in cellular metabolism, disruption of the plasma membrane, and alterations of lysosomal functioning, respectively. These assays were carried out on the same set of cells according to a modified version [27,30] of the protocol described by Dayeh et al. [41]. After the exposure (final volume of 200 µL/well) experiments, the medium was removed and cells were washed twice with 200 µL PBS. Wells received 100 µL of 1.25% (*v*/*v*) AB and 4 µM CFDA-AM prepared in serum-free/phenol red-free MEM (containing 1% NEAA). Fluorescence was measured on a Tecan Genios microplate reader (Tecan Group Ltd., Mannedorf, Switzerland) at a wavelength of 535/590 nm (excitation/emission) for AB, or at 485/535 nm for CFDA-AM after 30 min of incubation at 20 °C in the dark. Cells were washed once with 200 µL PBS and incubated with 100 µL of neutral red solution (0.03 mg/mL in serum-free/phenol red-free MEM containing 1% NEAA) for 1 h in the dark at 20 °C. Following incubation, cells were rinsed with PBS and the retained dye was extracted with 100 µL of an acidified solution (1% glacial acetic acid, 50% ethanol, and 49% Milli-Q water). Thereafter, fluorescence was measured at 532/680 nm (excitation/emission). The fluorescence values were corrected for the cell-free control results and normalized against the medium control values.

### 2.7. Generation of ROS

Cells were seeded and received the corresponding treatment in 96-well plates as described above. Cells were treated with increasing concentrations of GO and with chloramine-T trihydrate (0.04 mM–10 mM) used as positive controls. Intracellular ROS production was determined using the dichlorofluorescein (DCF) assay. After the exposure period, the medium and exposure compounds were removed, and cells were washed with PBS. Next, 100 μM of a 2′,7′-dichlorodihydrofluorescein diacetate (DCFH-DA) probe was added to each well. The plate was incubated at 20 °C in the dark for 30 min. After the incubation period, the DCFH-DA probe was removed, and the cells were washed twice with PBS. MEM phenol red-free medium was then added to the cells, and the fluorescence was measured at a 485 nm excitation and 535 nm emission using a Tecan Genios microplate reader. Fluorescence readings were taken immediately (time 0) and every 15 min over 60 min, with the plates maintained under dark conditions and incubated under exposure conditions between measurements. ROS production was calculated as the percentage increase in fluorescence per well over a 30-minute period using the formula [(Ft30 − Ft0)/Ft0 × 100], where Ft30 and Ft0 are the fluorescence measured at time 30 and 0 min, respectively. This result was finally expressed as a percentage of the control.

### 2.8. EROD and BFCOD Activities Measurement

Following exposure, the culture medium was aspirated, and the cells were rinsed with PBS. Subsequently, to measure EROD activity, 100 µL of 5 µM 7-ethoxyresorufin solution was added to each well and fluorescence was measured using a Tecan Genios microplate reader equipped with a 532 nm excitation and 590 nm emission filter. The fluorescence was read 11 times over 30 min, where the first measurement was carried out immediately after adding the probe and the last measurement after half an hour. BFCOD activity, based on benzyloxy-4-trifluoromethyl coumarin (BFC) metabolism into fluorescent HFC (7′-hidroxy-4 trifluoromethyl coumarin), was measured following the method described by Creusot et al. [37]. After exposure, the culture medium was removed and each well was rinsed once with PBS and refilled with 100 μL of PBS containing 40 μM of BFC. The kinetics of BFC metabolism by Cyp3A was then monitored in 10-minute intervals for 60 min in living cells with a spectrophotometer (Tecan Genios) using 405 and 532 nm as the excitation and emission wavelengths, respectively. Then, the medium was discarded and the total protein content in each well was determined using the fluorescamine method as previously described [42]. EROD and BFCOD activities were calculated as the quantity of resorufin or HFC (pmol, calculated using standard curves), respectively, generated in 1 min per mg of protein present in each well) (calculated using the fluorescamine assay [42]. Measurements were performed after four independent exposure experiments and in triplicate (three wells in each plate) for each individual sample.

### 2.9. Measurement of cyp1A and ahr mRNA Expression Levels by RT-qPCR

RTL-W1 cells were seeded in 6-well plates by adding 2 mL of a 2.5 × 10^5^ cells/mL cell suspension to each well. The plates were incubated at 20 °C for 24 h and then post-exposure experiments were performed. RTL-W1 cells were exposed to 0.01, 0.25, and 1 µM BkF for 24 h, then rinsed twice with PBS and post-exposed to GO at 18.75 µg/mL for 7 days. Cells exposed to the culture medium containing the corresponding percentage of DMSO (maximum 0.01%) and 2.5% H_2_O were used as vehicle controls.

Moreover, in order to confirm the mediation of the AhR in the enhancement of *cyp1A*, additional experiments with the AhR antagonist α-naphthoflavone (αNF) were performed. After a 24-hour seeding period, cells were pre-incubated with αNF 1 μM for 1 h before post-exposure experiments were carried out as described above. After the exposure periods, the cells were lysed, and the total RNA was extracted.

The total RNA was isolated using TRI Reagent (Ambion, Madrid, Spain) following the protocol indicated by the manufacturer. The concentration and purity (260/280 nm and 260/230 nm ratio) of the total RNA were determined by means of a Nano-Drop ND1000 spectrophotometer (Nano-Drop Technologies Inc., Wilmington, DE, USA).

Reverse transcription (RT) of 0.75 µg of isolated RNA was performed according to iScript™ cDNA Synthesis Kit (Bio-Rad, Madrid, Spain) using a PTC-100 programmable thermal controller (MJ Research Inc., Quebec, Canada). qPCRs (quantitative polymerase chain reaction) were performed using a Quantimix easy kit (Biotools, Madrid, Spain). Each reaction contained 2 µL of cDNA from RT (diluted previously four times), 10 µL 2X quantimix easy master mix, 2.2 µL of forward and reverse primer (diluted to 0.9 µM), and 3.6 µL nuclease-free H_2_O. The reaction protocol used for RT was the following: 5 min at 25 °C (priming), 20 min at 46 °C (reverse transcription), and 1 min at 95 °C (RT inactivation). qPCR was performed in an Applied Biosystems™ 7500 Fast Real-Time PCR System (Applied Biosystems, Waltham, MA, USA). Oligonucleotide primers for the amplification of *cyp1A*, *ahr*, and *elongation factor 1-alpha* (*eef1a*) as housekeeping gene cDNA are shown in Appendix A. The reaction protocol used for qPCR was the following: (1) 3 min at 95 °C (initial denaturation), followed by (2) 10 s at 95 °C (denaturation), followed by 1 min at 60 °C (annealing and amplification) (40 cycles), and (3) 1 min at 95 °C (1 cycle). Subsequently, the specificity of the qPCR was assessed by melting curve analysis (60–95 °C, increasing in 0.5 °C steps every 10 s). Each sample was loaded in duplicates. The relative expression levels of the target genes (*cyp1A* and *ahr*) were calculated using the ΔΔCt method. The housekeeping gene *eef1a* served as a reference gene to correct for differences in the total RNA input between samples.

### 2.10. Determination of Intracellular and Extracellular BkF Levels

In order to assess the role of BkF on Cyp1A induction, intra and extracellular levels of this chemical in culture wells were estimated by means of solid phase extraction (SPE) and high-performance liquid chromatography (HPLC) with fluorescence detection (FLD) analysis.

RTL-W1 cells were seeded in 6-well plates by adding 2 mL of a 2.5 × 10^5^ cells/mL cell suspension to each well. After 24 h at 20 °C, the cells were exposed to the highest concentration of BkF used in the study, 1 µM (252.31 µg/L), in a final volume of 2 mL, and cells were collected at different times between 1 h and 24 h (period corresponding to pre-exposure to BkF). Moreover, after 24 h of BkF treatment, the BkF internal and external cellular levels were also assessed from 24 h to 7 days (every 24 h), covering the post-exposure period. Cell culture medium was collected at the indicated sampling times and stored at −80 °C until BkF analysis. Cells were rinsed twice with PBS and frozen at −80 °C for 24 h, then lysed according to Myers et al. [43]. Briefly, 1 mL of ice-cold sucrose TKM buffer (sucrose 0.25 M, Tris 80 mM, KCl 25 mM, MgCl 25 mM, pH 7.4) was added to each well and the cells were scraped, collected in tubes, and kept on ice for 10 min. Cells were lysed by the addition of 20 μL 0.5% SDS for 2 min followed by an ultrasonic bath sonication for 10 min. This procedure was repeated a second time. BkF extraction from medium and cell lysates was performed by SPE using commercial cartridges Bond Elut C18 (50 mg, 1 mL) from Agilent Technologies (Folsom, CA, USA). The cartridges were preconditioned with 1 mL of methanol. After the conditioning step, 500 µL of medium or 1 mL of cell lysates were passed through the cartridges. The column was allowed to air dry, and analytes were eluted with 3 × 1 mL of methanol. Each volume was passed through the cartridge by gravity-controlled flow (no vacuum). The extracts from cells were passed through 0.45 µm filters before evaporation. The resulting extracts were vacuum-dried, reconstituted with 500 µL of methanol, and stored at −80 °C until injection into the HPLC system.

The presence of BkF in the samples was determined following the method of Gutiérrez-Valencia et al. [44] using an Infinity 1260 HPLC-System coupled to a fluorescence detector (Agilent Technologies Inc., Santa Clara, CA, USA). The chromatographic separation was achieved with a Poroshell 120SB-C18 column (Agilent Technologies Inc., Santa Clara, CA, USA), a mobile phase of methanol:water (98:2), and a flow rate of 0.4 mL/min. The fluorescence detector was set at an excitation wavelength of 263 nm and an emission wavelength of 410 nm. The linearity of the SPE–HPLC/FLD method was then evaluated. Calibration curves in medium and cell matrices (R2 = 0.99) in the range of 1–100 µg/L were used for the quantification of BkF. Precision was determined in terms of repeatability by injecting three times the same medium and cell extracts from one sample spiked with BkF at concentrations ranging between 1 and 100 µg/L. The relative standard deviations (RSDs) obtained showed values between 1.67% and 6.42% in the medium extract and between 2.64% and 7.81% in the cell extract. The efficiency of the recovery of BkF was assessed. For this purpose, 500 μL of medium were spiked with 5 µg/L or 90 µg/L of BkF. In the case of cells, 24 h after attachment, the medium was discarded, and the cells were frozen at −80 °C for 24 h. Then, cells were spiked with 500 µL of BkF at 5 µg/L or 90 µg/L in methanol. The extraction was performed as above. For each concentration, at least four replicates were made to determine the RSD. BkF recovery from the medium extracts was above the acceptable limit (>60%) as defined by the Association of Official Analytical Chemists (AOAC) International (2012) [45] for both tested concentrations, reaching 91 ± 8% for 5 µg/L and 80 ± 11% for the highest concentration (90 µg/L). BkF was not sufficiently recovered from the cellular matrix from either concentration, obtaining recoveries of 51 ± 10% and 56 ± 14% at 5 µg/L and 90 µg/L, respectively. However as shown, the repeatability (RSD) was acceptable at the two concentration levels tested. The limit of quantification (LOQ) for the BkF in the medium and cells was 1 µg/L.

### 2.11. Interference and Fluorescence Quenching

Before starting any experiment, the potential interferences, due to autofluorescence or to possible fluorescence quenching phenomena of GO suspensions, were assessed. Autofluorescence in the cell culture medium or when adhered to the cells after the treatments were measured by simulating the same conditions of the cytotoxicity assays, but without adding the corresponding fluorophores (AB, CFDA-AM, or NR). Non-treated cells served as a reference. Fluorescence quenching was tested in the presence of cells to simulate a more realistic assay scenario. For that, cells were seeded and exposed to GO (0.3–75 μg/mL) in the same way as for each assay. After the exposure period, the possible fluorescence quenching was assessed by incubating the exposed cells with the fluorophores that are formed in the course of each assay at the maximal concentration and at 10% of the maximal concentration that can be expected to be formed in the respective assays. Thus, for EROD and BFCOD activities, resorufin was prepared at 0.5 and 5 μM and HFC at 4 and 40 μM, respectively. For the cytotoxicity assays, resorufin (0.1 and 1 μM), 5-carboxyfluorescein (5-CF) (0.4 and 4 μM), and protonated NR (0.03 and 0.3 mg/mL) were prepared and used. For investigating fluorescence quenching in the ROS assay, DCF at 100 and 10 μM was used. Fluorescence readings of the exposed cells were taken at the same wavelengths used for each of the assays.

### 2.12. Statistical Analysis

Results are expressed as the mean ± standard error of the mean (SEM) of at least three independent experiments, in which each treatment was applied in duplicate or triplicate. Statistical analysis and graphical representations of the data were performed using GraphPad Prism 5 Software (San Diego, CA, USA). The normality of the distribution was confirmed with the Kolmogórov–Smirnov test. The homogeneity of variance was tested with Bartlett’s test. Data were compared between treatment and their corresponding control values using one-way repeated measurements analysis of variance (*p* < 0.05) followed by a post hoc Bonferroni’s test.

## 3. Results

### 3.1. Characterization of GO

The size frequency distribution of GO suspensions measured by DLS is shown in Table 1. GO stock suspensions (1 mg/mL) in Milli-Q water showed a main intensity peak of 275.81 ± 11.53 nm at time 0. This peak included 98.8% of the total distribution. The remaining 1.2% of the distribution corresponded to a peak of 4286.63 ± 554.7 nm. This distribution did not show variations either after 30 min of centrifugation or after 7 days (time of exposure), where 97.62 and 97.77% of the distribution gave 261.44 ± 15.39 nm and 261.24 ± 11.26 nm, respectively. A minor peak (2.38% and 2.23%) corresponding to a diameter of 4383.83 ± 168.45 nm and 4247.0 ± 292.61 nm, respectively, also remained present in these preparations. There was a tendency for GO to agglomerate in the L-15 media. Statistically significant differences (*p* < 0.001) were found between the size of the GO at the highest concentration (75 µg/mL) in the L-15 medium with respect to the GO stock suspension prepared in Milli-Q water, in the most abundant peak of the distribution (peak 1). Such differences are probably due to an agglomeration effect that was independent of time since the recorded particle size was similar at time 0 and 7 days. At lower concentrations (18.75 and 4.6 µg/mL), the agglomeration was lower, and statistically significant differences were not detected.

When analyzed by TEM, the size distribution of the GO suspensions (18.75 and 4.6 µg/mL) was in accordance with the DLS results. TEM images (Appendix A) revealed no remarkable differences in the size or shape between Milli-Q water (Appendix A) and L-15 media dispersions (Appendix A). Similarly, no differences were observed at different exposure durations, time 0 (Appendix A) and 7 days (Appendix A).

### 3.2. Interference of GO with Assays Components

No autofluorescence was detected for any GO suspensions in the culture medium or exposed cells under assay conditions.

For all fluorophores, a GO dose-dependent attenuation of fluorescence intensity was observed. However, at the highest concentration of GO suspension tested (75 µg/mL), the degree of quenching ranged from 5.67% to 17.76%, depending on the fluorophore (Appendix A). Results from these high exposure concentrations should be interpreted with caution, especially for the EROD, BFCOD, and AB. This has been taken into account in the interpretation of the results. The degree of quenching was independent of the fluorophore concentrations used in these assays.

### 3.3. Cytotoxicity of GO and Generation of Intracellular ROS

Results of the AB assay showed that GO significantly increased cellular metabolic activity after 7 days of exposure, reaching a maximum of 18.75 µg/mL (Figure 1A). The lowest observed effect concentration (LOEC) was 4.16 μg/mL according to the AB assay. At the highest GO concentration tested (75 µg/mL), the metabolic activity was reduced below control values. As this concentration of GO caused quenching (up to 17.76%), any drastic decreases in metabolic activity should be interpreted with caution.

The membrane integrity, assessed by means of the CFDA-AM assay, was significantly disrupted after 7 days of exposure to concentrations of 18.75 μg/mL (LOEC) (Figure 1A). The possible decrease in the fluorescence by quenching in this case was discarded (below 10%, even at the highest GO concentration tested).

In the NR assay, a significant decrease in fluorescence intensity was observed after 7 days of exposure to the highest GO concentration (LOEC of 100 μg/mL) (Figure 1A). In this case, the effect of quenching was also not contemplated.

Taking into account that the induction of oxidative stress is considered one of the principal mechanisms underlying nanomaterial toxicity [46,47], ROS generation was also assessed. GO induced intracellular ROS formation in a dose-dependent manner (Figure 1B) after 7 days of exposure. The ROS levels were found to be significantly (*p* < 0.05, and *p* < 0.001) elevated with respect to controls following exposure to concentrations ≥ 18.75 μg/mL (LOEC), reaching the maximal value of 584.1% at the highest concentration tested.

### 3.4. Interaction with the Plasma Membrane, Internalization, and Intracellular Fate of GO Observed by TEM

RTLW-1 cells were exposed to 4.16 μg/mL and 18.75 μg/mL for 7 days. Control cells observed by TEM exhibited normal structure without alterations (Figure 2A). In the TEM micrographs of ultrathin sections of GO-treated cells, numerous nanoplatelets were observed adjacent to the cell surface (Figure 2B–D). The micrographs demonstrated that GO nanoplatelets were able to come in close contact with the plasma membrane and penetrate through it, resulting in a disruption of the bilayer (Figure 2C,D). TEM images provided evidence that GO crossed the plasma membrane and accumulated inside the cell. GO was present as aggregate-like structures of different sizes and compactness and were either freely localized in the cytosol (Figure 2E) or enveloped within a membrane (Figure 2F).

### 3.5. Cyp1A and Cyp3A Induction after Single, Co, and Pre-Exposure to GO

Cells exposed to increasing concentrations of GO for 7 days did not show any induction of EROD or BFCOD activities. The co-exposure of cells to BkF and GO provoked an increase in the potency (estimated through the EC50) of the induction that was directly related to the concentration of GO used (EC50 of cells exposed to BkF alone was 0.126 µM and cells co-exposed to BkF and GO 18.75 µg/L was 0.02 µM) (Figure 3A). In the case of BFCOD activity, the co-exposure increased the potency of Cyp3A activation (EC50 value for BkF alone was 0.30 µM versus 0.02 µM in co-exposures with GO 18.75 µg/mL) (Figure 3B). However, the maximal value of the response after co-exposure with GO 18.75 µg/mL was lower (2.98 pmol/mg prot/min) than after the treatment with BkF alone (5.27 pmol/mg prot/min).

RTL-W1 cells were also pre-exposed to GO for 7 days and then (following removal of the nanomaterial-containing medium) incubated with increasing concentrations of BkF for 24 h. Again, RTL-W1 cell cultures pre-exposed to GO (18.75 µg/mL) exhibited higher maximal response in EROD activity (572.9 ± 81 pmol/mg prot/min) and lower EC50 values (0.002 µM) than cells incubated with BkF alone (474.4 pmol/mg prot/min and 0.006 µM) (Figure 3C). The effect of pre-exposure to GO on BFCOD activity was the inverse one, with a reduction in the maximal activity in cells pre-exposed to GO (5.2 pmol/mg prot/min) with respect to cells exposed to BkF alone (7.2 pmol/mg prot/min) (Figure 3D).

### 3.6. Cyp1A Induction after Post-Exposure to GO

RTL-W1 cells were treated with increasing concentrations of BkF for 24 h and, following removal of the BkF-containing medium, cells were washed with PBS and then incubated with GO (18.75 and 4.7 µg/mL) for 7 days, or with medium alone as a control. Under these conditions, cells post-exposed to 18.75 mg/mL of GO showed a notably higher EROD induction (171.19 pmol/mg prot/min) than cells incubated with BkF alone (68.53 pmol/mg prot/min) (Figure 4A). BFCOD activity was not induced in post-exposure conditions.

In addition, *cyp1A* and *ahr* mRNA levels were measured after 24 h of exposure with 0.01, 0.25, and 1 µM BkF and after post-exposure to GO (18.75 µg/mL) for 7 days. The results are shown in Figure 4B,C. Interestingly, GO was able to enhance *cyp1A* mRNA levels (3.17 times) even though it was not reflected in an increased EROD activity. BkF at a concentration of 1 µM caused a significant (*p* < 0.001) increase in *cyp1A* mRNA levels with respect to the controls (cells with the vehicle). RTL-W1 cells post-exposed to GO, after 24 h with 1 µM BkF, exhibited significantly higher (*p* < 0.001) *cyp1A* expression than cells exposed to GO alone or cells exposed to the different BkF concentrations alone. Similarly, cells exposed to BkF 1 µM exhibited a significant (*p* < 0.05) increase in *ahr* mRNA expression with respect to control cells (cells with vehicle). In cells post-exposed to GO, after 24 h with 1 µM BkF, such an increase with respect to the cells exposed to GO alone was higher since this increment was significantly different with respect to cells only exposed to BkF 1 µM (*p* < 0.05).

### 3.7. Effect of αNF on cyp1A and ahr Expression

In order to confirm the role played by the AhR on the observed responses under post-exposure conditions, a known AhR inhibitor, αNF, was used. The BkF concentration selected to perform this set of experiments was 0.25 µM. Results presented in Figure 5A show that GO significantly increased *cyp1A* mRNA expression, tripling it (*p* < 0.001). αNF alone did not cause any effect on *cyp1A* expression but was able to block the increment caused by GO. Again, post-exposure to GO for 7 days after 24 h with BkF 0.25 µM, significantly raised the *cyp1A* mRNA expression levels in cells, with respect to the effect caused after exposure with BkF alone and after post-exposure with GO for 7 days. In both cases, αNF abrogated the observed increases in *cyp1A* mRNA expression.

In the case of *ahr* mRNA expression (Figure 5B), neither GO nor αNF alone modified *ahr* gene transcription levels. Exposure of cells with BkF followed by post-exposure to GO for 7 days led to a significant increase in *ahr* gene expression with respect to both, exposure to BkF alone and to post-exposure with GO. Again, the observed *ahr* mRNA increase was significantly abrogated by αNF.

### 3.8. Generation of Intracellular ROS after Post-Exposure to GO

In order to determine whether the Cyp1A over-induction observed in post-exposure conditions could be due to an increase in ROS, intracellular ROS levels were determined. The results presented in Figure 6 show that ROS were not increased in a BkF concentration-dependent manner (under BkF exposure for 24 h, followed by 7 days with media only). However, cells exposed to GO alone and cells post-exposed to GO after exposure to a range of BkF concentrations for 24 h, showed a significant increase in ROS levels, which were not BkF dose-dependent.

### 3.9. Role of BkF on the Over-Induction of the Cyp1A after Post-Exposure to GO

The internal and extracellular levels of BkF were assessed by means of SPE-HPLC-FLD, in order to discard that the effects observed on the Cyp1A induction after the post-exposure conditions could be due to the BkF remaining in the culture medium after 24 h of treatment. The BkF concentration decreased in the extracellular medium from 1 h onwards while intracellular BkF increased during the first 7 h of treatment to decrease thereafter (Appendix A). After BkF withdrawal (24 h), no levels of this compound could be detected in the medium or the cellular lysates.

## 4. Discussion

Due to their unique characteristics and extensive applications, the toxic potential of GO materials has received maximal attention in recent years [15,20,21,39,48]. However, very little is known about the mechanisms underlying its possible metabolism or bioaccumulation in organisms. Additionally, if released into ecosystems, GO will interact with other pollutants, leading to mutual modulations of their toxicity, although these aspects are not yet fully understood [10,22,49]. In the present work, we used an environmental pollutant, BkF, to first induce detoxification activities in a rainbow trout liver-derived cell line and observed that exposure to GO after retiring BkF (post-exposure conditions) led to a strong potentiation of such activities; in particular, of those related with the Cyp1A system. Together with Cyp1A, Cyp3A induction was assessed, only for comparative purposes since Cyp3A activation is not dependent on the AhR.

The study of the interactions of GO and BkF evidenced the mutual influence of both substances on their toxicity but, what is more important, gave essential clues about the possible role of the AhR and Cyp1A on the cellular metabolism of GO. The obtained information has important implications for GO risk assessment at a regulatory level since it suggests that GO could be metabolically removed from organisms instead of simply bioaccumulated.

Studies were performed using RTL-W1 cell cultures, an in vitro system that has proven to be useful in multiple nanomaterial toxicity studies [28,29,50,51,52,53]. These cells have been recently proposed to perform long-term cytotoxicity studies [26] since they show stable monolayer cultures without requiring trypsinization or medium changes for seven days. A comprehensive characterization of the GO physicochemical properties under culture conditions along the exposure time was performed. The DLS measurements showed that the stock suspensions of the GO used in the present work were stable in Milli-Q water for up to 7 days at 20 °C. Controlled toxicity testing requires the use of homogeneous test samples, but obtaining stable dispersions without influencing their toxicological behavior is one of the major challenges in in vitro toxicology [54]. In the present study, the HDD of GO used was 275.81 ± 11.53 nm, similar to that used in other studies where the toxicity of fish cells was analyzed, ranging, in these cases, from 100 to 382 nm [25,30,55]. The DLS measurements evidenced the tendency of GO to agglomerate in L-15 media at the higher GO concentrations tested. Therefore, all effects caused at 75 µg/mL should be interpreted with caution. At the lower concentrations tested (4.6 and 18.75 µg/mL), GO agglomeration decreased and the measured sizes were similar to those obtained in Milli-Q water. These results proved that the dispersion protocol used [30] was adequate to obtain stable dispersions along the exposure time (7 days) at the applied GO concentrations.

TEM images showed the internalization of GO into RTL-W1 cells. It was observed that GO could penetrate through the plasma membrane, causing the disruption of the bilayer. Previous experimental and computer simulation studies have demonstrated that the cellular uptake of GO is initiated at the corners or edges of the sheets in order to overcome high energy barriers [56]. In the present study, no specific markers or inhibitors were applied to elucidate the GO cellular uptake pathways, however, their intracellular localization in membrane-enclosed compartments suggests an endocytic uptake process. Similar results were also obtained by Kalman et al. [25] for GO. On the other hand, as described for carbon nanotubes [57,58], GO could be able to spontaneously translocate into the cytosol after disrupting the plasma membrane by means of a mechanism resembling a “needle-like” entry, appearing then freely in the cytosol. This needle-like mechanism was also previously suggested for GO uptake [27,30]. In contrast, a subsequent recent study with primary rainbow trout hepatocytes showed that the same GO used in our study was not internalized by the cells, but it was able to affect cellular metabolic activity and cell membrane integrity [59]. The uptake process of GO would need further in-depth investigation considering differences among GOs and cellular models.

The initial cytotoxicity assays provided a means to select the appropriate concentrations to study Cyp induction. The LOEC results in this study ranged from 4.16 to 100 µg/mL depending on the assay used. Previous studies showed similar results, with LOEC values ranging from 0.125 to 16 µg/mL depending on the cell lines used, the cytotoxic assay employed, and the GO material tested [21]. Taking into account the cytotoxicity results, GO concentrations of 4.6 and 18.75 µg/mL were selected as they were not toxic to RTLW-1 cells. At 18.75 µg/mL, GO was able to increase the metabolic activity (according to the AB assay), provoke plasma membrane damage, and cause significant ROS enhancement. On the other hand, the concentration of 4.6 µg/mL was selected since it did not have any of the effects mentioned above.

In order to assess the influence of GO on the detoxification activities induced by BkF, a 7-day exposure period was selected. BkF is an AhR ligand and is considered one of the most potent Cyp1A inducers [27,60]. Additionally, BkF is also able to activate Cyp3A [37]. Therefore, BkF was selected as a reference environmental pollutant, which is able to induce both Cyp1A and Cyp3A detoxification systems, allowing a direct comparison of the induction process of both systems to give clues about their role in the metabolism of GO.

As it was expected, BkF alone led to an increase in EROD and BFCOD activities that were directly dependent on the exposure concentration. Exposure to GO alone did not lead to the induction of EROD or BFCOD activities. In the case of EROD activity, it was observed that the potency of BkF to induce this activity clearly increased in co-exposure experiments with the concentration of GO. Similar results were obtained by Lammel et al. [24] in relation to Cyp1A induction in PLHC cells after co-exposure to a different GO and BkF or other AhR ligands. In the mentioned study, Cyp1A induction was associated with GO’s ability to increase the intracellular concentration of BkF, through enhanced uptake via a carrier-like mechanism or just by destabilizing the membrane integrity and allowing the passive diffusion of BkF to the inside of the cells [24]. GO has been described as an adsorbent of co-existing conventional pollutants in water due to its large surface area and abundant functional oxygen groups [11]. In the present study, the chemical analysis of BkF showed that BkF was internalized and metabolized in the first 24 h. However, the results of the co-exposure experiments did not allow for discarding the possibility that GO could effectively be acting as a carrier and contributing in such a way to the observed Cyp1A induction. In the case of BFCOD activity, the co-exposure experiments revealed an increase in the potency of BkF to induce this activity but a parallel reduction in the amplitude of the response was observed. Recently, it has been reported that GO could be able to inhibit the Cyp3A4 catalytic activity in microsomal-based models as well as its gene expression in HepG2 and HepaRG cell lines [61]. If GO acts as a carrier, BkF can activate Cyp3A faster than in the absence of GO. However, on the other hand, GO could also block Cyp3A, establishing a competition between BkF and GO and making the maximum response lower than this seen with BkF alone, as observed in the present work. Globally, the obtained results reveal a clear effect of GO on the Cyp1A system but possibly a limited direct influence on Cyp3A, although additional studies would be needed to assess in depth the role of GO on Cyp3A activation.

The enhanced effect of GO on Cyp1A and Cyp3A could be caused by other mechanisms distinct from a carrier effect. Two additional possibilities have been addressed in this study; the first one was that GO could facilitate the passive diffusion of BkF into the cytosol due to graphene-induced damage or destabilization of the plasma membrane, as mentioned above. To investigate this, RTL-W1 cells were first pre-exposed to a fixed concentration of GO for 7 days and then incubated with increasing concentrations of BkF for 24 h. Interestingly, the EROD dose-response curves of the pre-exposure experiments resembled those of the co-exposure experiments, demonstrating that the observed induction of Cyp1A must not be necessarily associated with an enhancement of the internalization of BkF caused by GO. However, the effect on Cyp3A was the inverse: the BFCOD activity showed a decrease in the maximal response, which could support the hypothesis raised before where the GO would act by blocking the Cyp3A.

Lastly, we hypothesized that given the polyaromatic nature of graphene it is possible that after an initial degradation of GO, intermediate metabolites appear exhibiting structural similarities to PAHs that could cause the induction of Cyp1A through molecular mechanisms similar to those caused by other Cyp1A inducers. For instance, it has been observed previously that peroxidases could participate effectively in the degradation of graphene so that small fragments with a polyaromatic nature could be released [62,63]. To test if the formation of graphene-derived metabolites with AhR agonistic activity could explain the increase in Cyp1A activity observed in the co-exposures, RTL-W1 cells were first pre-incubated with BkF (24 h) to induce the cellular machinery involved in xenobiotic metabolism and then exposed to GO (7 days). The results showed an enhanced induction of Cyp1A when cells were post-exposed at 18.75 µg/mL of GO after 24 h with BkF. The effect was more pronounced than in the co-exposure experiments. Surprisingly, although GO alone was not able to induce EROD activity, at the transcriptional level, the *cyp1A* mRNA was over-expressed upon GO exposure for 7 days. This effect was even stronger when the AhR machinery was activated by BkF in post-exposure conditions. Our results are consistent with previous studies performed in zebrafish, where graphene derivatives up-regulated P450 family genes, suggesting that the used graphene-related materials were effectively metabolized by P450 monooxygenases via the canonical AhR signaling pathway [31,32,33]. Lammel et al. [24], in a similar study, were not able to observe this potentiating effect in PLHC-1 cells, however, shorter-term (24 h) exposures were used and this could explain the lack of the effect. To corroborate the mentioned ideas about a possible effect of GO fragments and taking into account that *cyp1A* mRNA enhancement should occur after AhR activation by metabolites, a known AhR inhibitor, αNF, was used to identify the direct involvement of the AhR in the effect. It was observed that αNF abrogated the effect caused by GO on *cyp1A* and *ahr* mRNA expression, supporting the hypothesis that GO metabolites could be responsible for AhR activation.

The role of ROS in the enhanced induction of Cyyp1A after post-exposure to GO was also assessed. ROS have been reported to be generated during NADPH-dependent drug metabolism by constitutive Cyps in liver microsomes [64], and also in isolated hepatocytes [65,66], so that induction of ROS could be parallel to the observed Cyp induction. In the present work, ROS were measured after post-exposure conditions (24 h BkF, then 7 days with GO) when Cyp1A enhancement was maximal. Because ROS were determinant in the induction of EROD activity and Cyp1A after post-exposure to GO, an increment in ROS levels could be expected after pre-exposure conditions parallel to EROD activity induction. However, the results show the opposite, with a clear decrease in ROS at the highest BkF concentrations.

Lastly, in order to discard that the effects observed on the Cyp1A induction under post-exposure conditions could be due to the remains of BkF present in the medium or cells after treatment, the levels of this compound were determined by means of SPE-HPLC-FLD both, in the cell culture medium and extracts of cultured cells. The results showed that BkF was undetectable after 24 h of exposure in either case. This discarded the hypothesis that the enhanced induction of Cyp1A could be due to the presence of BkF in the cells. In addition, possible effects due to BkF metabolites can be ruled out as the addition of medium without GO did not induce EROD activity (Figure 4A).

## 5. Conclusions

This study gives evidence of a possible potentiating effect of GO on PAH-induced Cyp1A expression at both the transcriptional and enzymatic levels. The results further suggest that the increase in Cyp1A induction observed under the post-exposure conditions, where the detoxification machinery is active, could be related to an active metabolization of GO mediated by the activation of AhR and induction of detoxification-related activities, including those dependent on Cyp1A. The obtained results also indicate that preceding, simultaneous, and post-exposure to GO in polluted environments may modify the toxicokinetics of aromatic environmental pollutants such as PAH. These discoveries demonstrate that the combination effects between GO and environmental pollutants have to be considered when evaluating their respective hazard. Further studies are needed to provide a detailed description of the underlying molecular mechanisms and signal transduction pathways as well as to offer a more comprehensive toxicity assessment of GO not only in fish but in different aquatic organisms.

## Figures and Tables

**Figure 1 nanomaterials-13-02501-f001:**
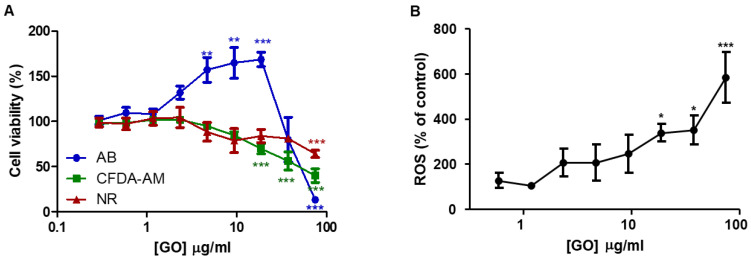
Effect of graphene oxide (GO) on RTL-W1 cell viability. (**A**) RTL-W1 cells were exposed to increasing concentrations of GO for 7 days. The cytotoxicity of GO was assessed by means of the AB assay, 5′CFDA-AM assay, and NRU assay. (**B**) Level of intracellular reactive oxygen species (ROS) upon exposure to increasing concentrations of GO for 7 days. The data points and error bars represent the mean and standard error of the mean (SEM) of at least three independent experiments. Statistically significant differences with respect to the vehicle control (one-way rm ANOVA, Dunnett’s post hoc test) are indicated as follows: * *p* < 0.05, ** *p* < 0.01, and *** *p* < 0.001.

**Figure 2 nanomaterials-13-02501-f002:**
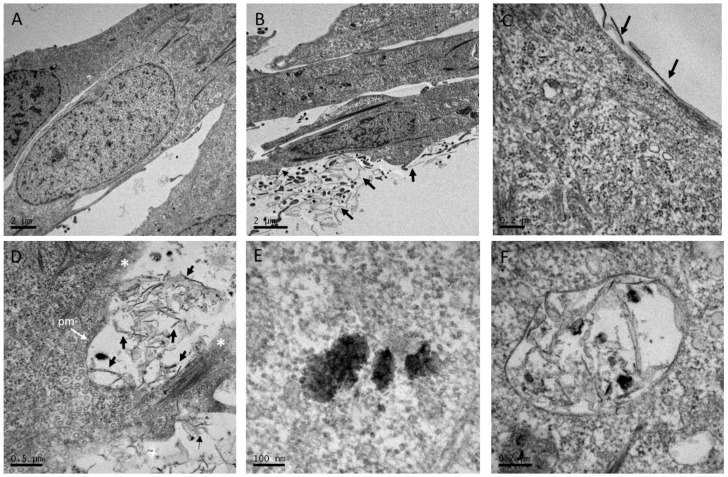
Selected TEM images of RTL-W1 non-exposed cells (**A**) and 7 days GO exposed at 18.75 µg/mL (**B**–**F**). (**B**,**C**) GO nanoplatelets (black arrows) observed adjacent to the cell surface. (**C**,**D**) GO nanoplatelets (black arrows) interacting with the plasma membrane (pm) and penetrating the latter, leading to plasma membrane disruption (the site of disruption is indicated with a white asterisk). (**E**,**F**) Intracellular aggregation of GO nanoplatelets (black arrows). Intracellular location of GO throughout the cytoplasm (**E**) or in membrane-surrounded vesicles (**F**).

**Figure 3 nanomaterials-13-02501-f003:**
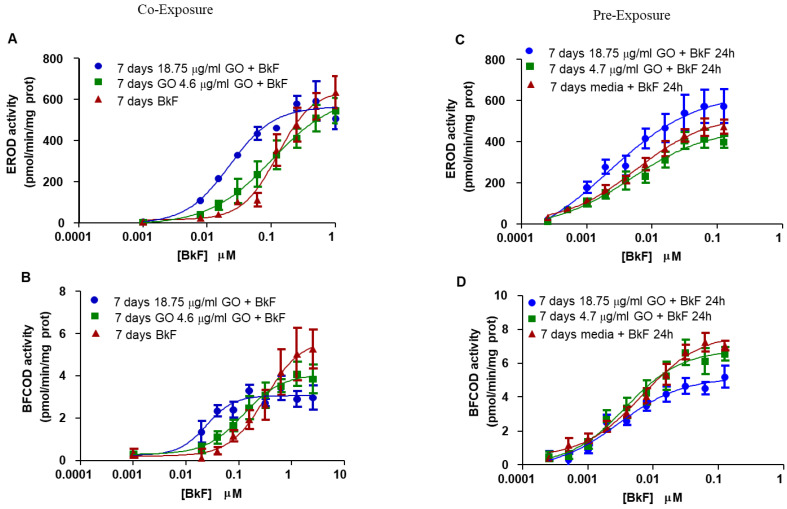
Ethoxyresorufin-*O*-deethylase (EROD) and Benzyloxy-4-trifluoromethylcoumarin-*O*-debenzyloxylase (BFCOD) activities after co-exposure (**A**,**B**) and pre-exposure (**C**,**D**) to GO and BkF. (**A**,**B**) RTL-W1 cells were co-exposed to increasing concentrations of Benzo(k)fluoranthene (BkF) (0.001–1 µM) alone or together with 4.6 µg/mL GO or 18.75 µg/mL GO for 7 days. (**C**,**D**) cells were pre-exposed to GO at 4.6 µg/mL GO or 18.75 µg/mL GO for 7 days and then to increasing concentrations of BkF (0.00025–0.125 µM) alone for 24 h. The EROD activity levels measured in cell cultures co-exposed and pre-exposed to GO and BkF are shown in (**A**,**C**), respectively, and represented as pmol of resorufin/min/mg protein. The BFCOD activity levels measured in cell cultures co-exposed and pre-exposed to GO and BkF are shown in (**B**,**D**), respectively, and represented as pmol of HFC/min/mg protein. The data points and error bars represent the mean and standard error of the mean (SEM) of at least three independent experiments.

**Figure 4 nanomaterials-13-02501-f004:**
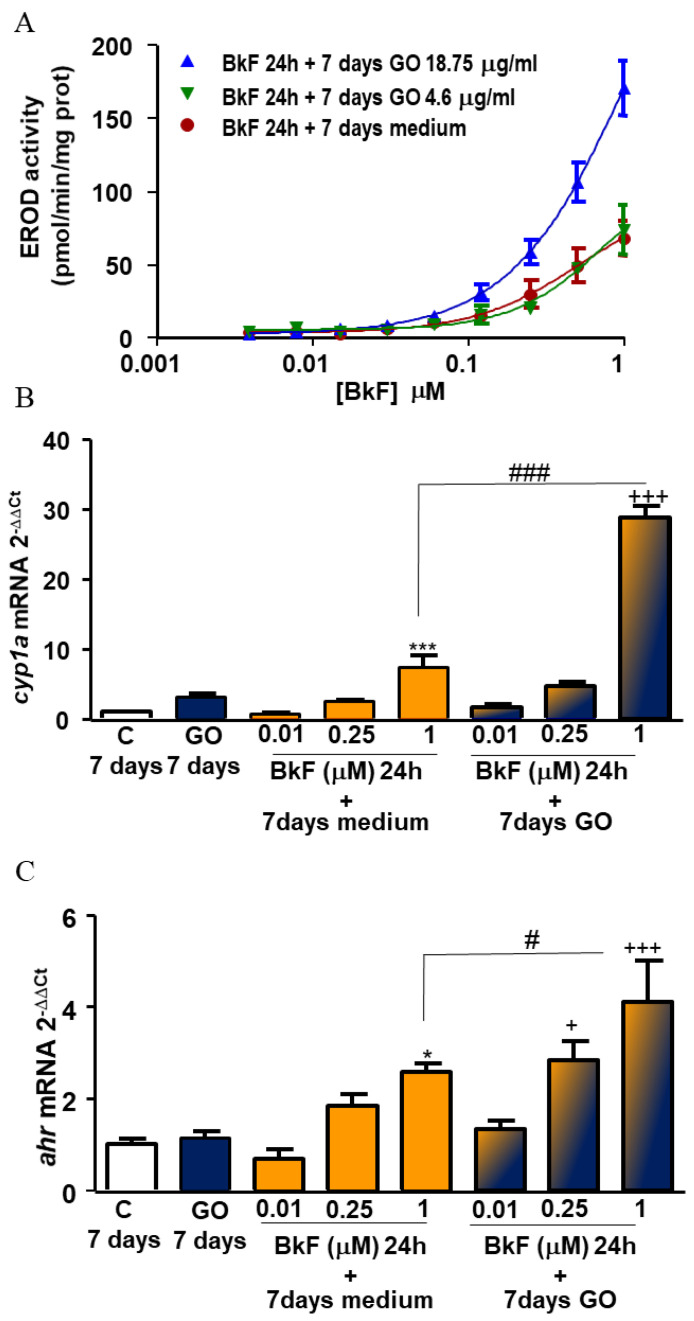
EROD activity and mRNA expression after post-exposure to GO. (**A**) EROD activity levels in RTL-W1 cells pre-incubated with increasing concentrations of BkF (0.01–1 µM) for 24 h and post-exposed to media, 4.6 µg/mL, or 18.75 µg/mL of GO for 7 days. The EROD activity levels are represented as pmol/min/mg protein. *cyp1A* (**B**) and *ahr* (**C**) mRNA expression levels in RTL-W1 cells pre-incubated with 0.01, 0.25, and 1 µM of BkF for 24 h and post-exposed to media or to 18.75 mg/mL GO for 7 days. *cyp1A* and *ahr* mRNA expression are represented as a fold expression with respect to the corresponding mRNA expression level in control cells. Statistically significant differences with respect to the control cells (vehicle) are indicated as * for *p* < 0.05 and *** for *p* < 0.001 and with respect to GO 7 days as + for *p* < 0.05 and +++ for *p* < 0.001. Statistically significant differences between mRNA *cyp1A* levels measured for the same BkF concentration with and without post-exposure to GO 18.75 mg/mL for 7 days are indicated as # and ### (*p* < 0.05 and <0.001) (One-way rm ANOVA, Bonferroni’s Multiple Comparison test). Data points and error bars represent the mean and standard error of the mean (SEM) of at least three independent experiments.

**Figure 5 nanomaterials-13-02501-f005:**
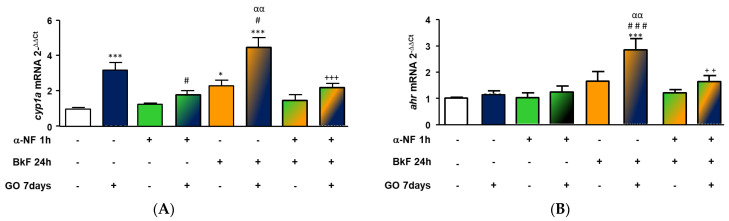
*cyp1A* (**A**) and *ahr* (**B**) mRNA expression levels. RTL-W1 cells were pre-incubated for 1 h with 1 µM of α-Naphtoflavone (αNF), then exposed to BkF (0.25 µM) for 24 h more, and finally post-exposed to GO 18.75 µg/mL for 7 days. mRNA expression levels are represented as a fold expression with respect to the levels measured in the control cells. Statistically significant differences with respect to the vehicle control are indicated as: * for *p* < 0.05 and *** for *p* < 0.001; with respect to GO 7 days as # for *p* < 0.05 and ### for *p* < 0.001; with respect to BkF 0.25 µM for 24 h then 7 days with media as αα for *p* < 0.01; and with respect to BkF 0.25 µM for 24 h then 7 days with GO 18.75 mg/mL as ++ for *p* < 0.01 and +++ for *p* < 0.001 (One-way rm ANOVA, Bonferroni’s Multiple Comparison test). Bars and error bars represent the mean and standard error of the mean (SEM) of three independent experiments.

**Figure 6 nanomaterials-13-02501-f006:**
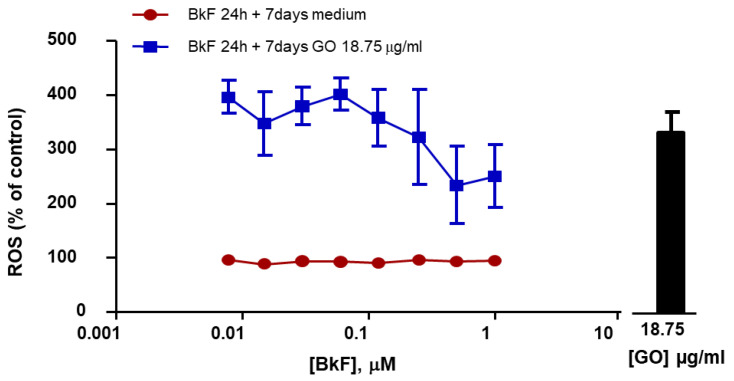
Levels of intracellular reactive oxygen species (ROS) upon post-exposure conditions to GO. RTL-W1 cells were pre-incubated with increasing concentrations of BkF (0.078–1 µM) for 24 h and post-exposed 7 days with media or with GO at 18.75 µg/mL. The bar represents the value of ROS after exposure to GO at 18.75 µg/mL alone. Point, bar, and errors represent the mean and standard error of the mean (SEM) of three independent experiments.

**Table 1 nanomaterials-13-02501-t001:** Hydrodynamic size distribution of GO.

Sample	Time (Days)	Z-Ave (nm)	PDI	Average HDD, nm ± SEM (Peak % with Respect to Total Intensity)		
				Peak 1	Peak 2	Peak 3
L15 medium (L15)	0	14.8 ± 0.34	0.4 ± 0.01	14.86 ± 1.13 (63.97)	61.42 ± 1.30 (35.00)	4223.25 ± 89.59 (1.03)
GO stock dispersion	0	224.8 ± 9.59	0.24 ± 0.03	275.81 ± 11.53 (98.8)	4286.63 ± 554.7 (1.2)	
GO stock dispersion after centrifugation	0	204.4 ± 9.64	0.24 ± 0.01	261.44 ± 15.39 (97.62)	4383.83 ± 168.45 (2.38)	
GO stock dispersion after centrifugation	7	198.0 ± 6.97	0.25 ± 0.01	261.24 ± 11.26 (97.77)	4247.00 ± 292.61 (2.23)	
GO (75 µg/mL)-L15	0	328.6 ± 26.82	0.43 ± 0.02	514.40 ± 19.15 *** (85.02)	4262.68 ± 181.92 (6.53)	112.84 ± 11.75 (8.45)
GO (75 µg/mL)-L15	7	385.2 ± 42.48	0.5 ± 0.02	572.00 ± 44.82 *** (83.62)	4668.21 ± 170.16 (7.7)	137.15 ± 14.62 (8.68)
GO (18.75 µg/mL)-L15	0	182.3 ± 3.89	0.40 ± 0.06	283.33 ± 18.71 (93.89)	31.56 ± 4.83 (3.91)	4753.92 ± 46.35 (2.2)
GO (18.75 µg/mL)-L15	7	234.5 ± 20.31	0.45 ± 0.06	348.70 ± 18.27 (89.94)	64.07 ± 6.38 (7.9)	4530.82 ± 141.83 (2.16)
GO (4.6 µg/mL)-L15	0	157.1 ± 37.78	0.63 ± 0.02	328.9 ± 59.26 (86.17)	33.64 ± 5.88 (8.96)	8.96 ± 1.3 (4.21)
GO (4.6 µg/mL)-L15	7	155.88 ± 19.19	0.64 ± 0.04	344.10 ± 39.48 (85.57)	31.13 ± 7.7 (9.62)	6.61 ± 0.01 (4.81)

Z-average (z-ave), polydispersity index (PdI), average hydrodynamic diameter (HDD), and percentage of each peak with respect to the total intensity (%). Mean ± SEM (*n* = 3–10). Statistically significant differences with respect to the GO stock dispersion after centrifugation (one-way rm ANOVA, Dunnett’s post hoc test) are indicated as *** *p* < 0.001.

## Data Availability

The data presented in this study are available on request from the corresponding author. The data are not publicly available due to limitations in the availability of data until the end of the project. Thereafter, data presented in this study will be openly available in Digital.csic at 10.20350/digitalCSIC/13518.

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
