# Peer review of "The Potentiating Effect of Graphene Oxide on the Arylhydrocarbon Receptor (AhR)–Cytochrome P4501A (Cyp1A) System Activated by Benzo(k)fluoranthene (BkF) in Rainbow Trout Cell Line"

_nanomaterials, 2023, doi:10.3390/nano13182501_

Round 1

Reviewer 1 Report

  • Authors  investigated if graphene oxide (GO) could be metabolized by the aryl hydrocarbon receptor (AhR) and cytochrome P4501A (Cyp1A). Using rainbow trout cells, the authors activated AhR with benzo(k)fluoranthene. In later experiments, GO strongly induced Cyp1A even without benzo, suggesting GO activates AhR. Co-treatment with an AhR antagonist blocked this, confirming AhR's role. Results indicate that after initial degradation, GO fragments may be further metabolized by AhR and Cyp1A. This sheds light on the potential environmental fate and metabolism of GO. The study suggests GO could modulate toxicity of other AhR agonists. Here are some of the comments/ concerns/ suggestions:

    1.     GO was commercially purchased but batch to batch variations and knowing impurities could shred more information of the GO effect in the in vitro testing. Adding  Raman and IR spectroscopy information could add some vital information.

    2.     How do the concentrations where GO caused toxicity in cells stack up to other studies? Are the thresholds a lot different or pretty similar?

    1. The paper talks about quenching effects messing up fluorescent assays for GO. Did they do stuff to try and fix that problem? Not totally sure they really accounted for it.
    2. The techniques they used looked at how GO nanoparticles clump together in media. But did they really test if that changed a lot across the concentrations they tested? Seemed kind of lacking.
    3. The way GO got inside cells in the images looks like what other groups have published. But could there be other reasons to explain what they saw with the TEM pictures?  unclear.
    4. They say GO alters some CYP enzymes but different ones and even opposite effects compared to other papers. Why would that be? Doesn't really make sense to me.
    5. They checked that the BkF chemical was cleared out of the cells after treatement. But does that match up with pharmokinetic data on how fast it should be eliminated? Maybe some was still around to influence things.
    6. How similar was the GO they used to stuff from other studies - size, surface coatings etc? No clue if its really comparable GO.
    7. Are the doses of GO and BkF realistic based on human exposures? If its super high may not translate clinically.

Reviewer 2 Report

Valdehita and colleagues have presented compelling evidence indicating a potential enhancement effect of graphene oxide (GO) on PAH-induced Cyp1A expression, manifesting at both the transcriptional and enzymatic levels. Their findings suggest that the augmented Cyp1A induction observed in post-exposure conditions, characterized by active detoxification machinery, could be attributed to the active metabolization of GO. This process appears to be mediated by the activation of AhR and the subsequent induction of detoxification-related activities, including those dependent on Cyp1A. Importantly, the study's outcomes also propose that sequential exposures to GO, whether preceding, simultaneous, or post-exposure, within polluted environments, may induce modifications in the toxicokinetics of aromatic environmental pollutants like PAH. These discoveries underscore the necessity of considering combination effects between GO and environmental pollutants when assessing their individual hazards. The research, as presented, holds substantial significance for the readership of Nanomaterials. Nevertheless, prior to publication, the authors are encouraged to address certain critical aspects to fortify the credibility and robustness of their study:

1.      Although the study offers indications of GO's augmentation of the AhR-Cyp1A system, a more comprehensive toxicity assessment is imperative. Ensuring the absence of adverse effects on ecological systems and human health in practical applications is paramount.

2.      While the study has shed light on the interaction between oxidized graphene and the AhR-Cyp1A system, a more detailed investigation into the underlying molecular mechanisms and signal transduction pathways is warranted to establish a deeper understanding.

3.      The potential interactions of oxidized graphene with other environmental factors in real-world settings are of utmost concern. Conducting further research to evaluate the behavioral outcomes and broader impacts within authentic ecological systems is recommended.

4.      The interaction dynamics and biological accessibility of oxidized graphene to organisms represent a crucial dimension. Therefore, a thorough comprehension of its behavior and potential risks within biological entities is crucial.

Minor editing of English language required

Round 2

Reviewer 1 Report

The authors have addressed all the concerns raised. Congrats on the great scientific contributions